# Improved real-time bio-aerosol classification using Artificial Neural Networks

**Maciej Leśkiewicz[1], \*Miron Kaliszewski[2], Maksymilian Włodarski[2], Jarosław Młyńczak[2], Zygmunt Mierczyk[2], Krzysztof Kopczyński[2].**

1. PCO S.A. ul. Jana Nowaka-Jeziorańskiego 28, 03-982 Warsaw, Poland.
2. Institute of Optoelectronics, Military University of Technology, ul. Gen. Witolda Urbanowicza 2, 00-908 Warsaw, Poland

\*Corresponding author: miron.kaliszewski@wat.edu.pl

**Keywords: Bio-aerosol, Fluorescence, Real-time analysis, Artificial Neural Network, PBAP.**

## 1. Abstract

Air pollution has had an increasingly powerful impact on the everyday life of humans. Ever more people are aware of the health problems that may result from inhaling air which contains dust, bacteria, pollens or fungi. There is a need for real-time information about ambient particulate matter. Devices currently available on the market can detect some particles in the air but cannot classify them according to health threats. Fortunately, a new type of technology is emerging as a promising solution.

Laser based bio-detectors are opening a new era in aerosol research. They are capable of characterizing a great number of individual particles in seconds by analyzing optical scattering and fluorescence characteristics. In this study we demonstrate the application of Artificial Neural Networks (ANNs) to real-time analysis of single particle fluorescence fingerprints acquired using BARDet (a Bio-AeRosol Detector). 48 different aerosols including pollens, bacteria, fungi, spores, and non-biological substances were characterized. An entirely new approach to data analysis using a decision tree comprising 22 independent neural networks was discussed. Applying confusion matrices and ROC analysis the best sets of ANNs for each group of similar aerosols was determined. As a result,  a very high accuracy of aerosol classification in real-time was achieved. It was found that for some substances that have characteristic spectra almost each particle can be properly classified. Aerosols with similar spectral characteristics can be classified specific clouds with high probability. In both cases the system recognized aerosol type with no mistakes.

In the future, it is planned that performance of the system may be determined under real environmental conditions, involving characterization of fluorescent and non-fluorescent particles.

## 2. Introduction

Ambient air contains a variety of particles such as dust, bacteria, pollens, fungi and other particles of biological and non-biological origin (Pöhlker et al., 2013; Górny, 2004). Aerosols are involved in various atmospheric processessuch as ice nuclei formation, precipitation and global climate effects (Deguillaume et al., 2008; Fröhlich-Nowoisky et al., 2016; Gabey et al., 2010; Pósfai and Buseck, 2010; Fuzzi et al., 2015). They also greatly influence human health (Davidson et al., 2005; Pope and Dockery, 2006; Michaels, 2017; Shiraiwa et al., 2012). Therefore, the characterization of ambient air is important for estimating potential health hazards and environmental impact (Mauderly and Chow, 2008; Lim et al., 2005). Standard methods of aerosol composition assessment usually include microscopic inspection or molecular analysis of filters (Miaskiewicz-Peska and Lebkowska, 2012), tape or liquid trapped particles. Nevertheless, they suffer from low time

resolution due to periodical and relatively long analytical procedures. They are also ineffective for the
detection of non-culturable microorganisms (Blais-Lecours et al., 2015; Trafny et al., 2014).
The detection and classification of biological particles is possible using fluorescence techniques
due to the presence of proteins, NADH, and some vitamins that emit light when excited with UV light
(Lakowicz, 2006). This feature is utilized in single particle fluorescence detectors. In the flowing air
each particle is characterized for size/shape using light scattering as well as fluorescence properties.
This approach ensures continuous measurement and immediate response. Thus the analysis process
can be facilitated and accelerated compared with other commonly used analytical procedures (Hill et
al., 1999; Choi et al., 2014; Taketani et al., 2013; Feugnet et al., 2008). Besides advantages such as
reagentless and real time particle characterization, the laser based methods do not provide
information on the chemical composition of aerosol.
Several studies using single particle fluorescence detectors have demonstrated that fluctuations
of aerosol concentration and variations in its fluorescence properties are highly dependent on the
season, day, time, location and place occupancy (Gabey et al., 2011; Huffman et al., 2010; Pinnick et
al., 2004; Bhangar et al., 2014; Fennelly et al., 2017). Each single particle passing the instrument is
labelled with a time stamp, scattering properties (size and/or shape) and fluorescence
characteristics. It is obvious that continuous single particle measurements bring a new potential and
quality to environmental research. However, particles of the same type and batch display slightly
different spectral characteristics due to variations in biochemical composition, size, age of population
(Agranovski et al., 2003), degradation (Hernandez et al., 2016) or stress level (Lee et al., 2010) and
the particle position within the instrument's interrogation point (Pan et al., 2011). Simpler statistical
analyses, such as data averaging and graphical spectra representation, are not sufficient. Therefore,
the huge amount of data and occurring spectral variations require more advanced algorithms
supporting automatic data classification. Various analytical methods of particle discrimination and
classification have been applied. It has been shown that Principal Component Analysis (PCA), Linear
Discriminant Analysis (LDA), Hierarchical cluster Analysis (HCA) of fluorescence spectra greatly
increase discrimination of particles compared with methods based on spectra averaging or
fluorescence threshold (Leśkiewicz et al., 2016; Kaliszewski et al., 2013; Pan et al., 2012; Savage et al.,
2017; Crawford et al., 2015). Artificial neural networks (ANNs) comprise an emerging analytical
approach that is becomeing more widely and successfully applied in various life domains such as
chemical analysis (Borecki et al., 2008), image recognition (Antowiak and Chałasińska-Macukow,
2003), data mining and weather forecasting (Purnomo et al., 2017). It has been shown that ANNs can
be applied in bio-aerosol classification (Kohlus and Bottlinger, 1993). However, it usually requires
more user input compared to other analytical procedures (Ruske et al., 2017).
This paper focuses on the application of ANNs for real time discrimination of bio-aerosols based
on single particle fluorescence characteristics. We demonstrate a new approach to data analysis
using ANNs which allows automation of data preparation procedures and minimum user
involvement.
**3.   Materials and methods**
**3.1.  Experiment**
**3.1.1.   BioAeRosol Detector (BARDet)**
Detailed information concerning the construction and parameters of the instrument used for
the experiments was presented in our previous work (Kaliszewski et al., 2016). In general, the
ambient air is continuously drawn through the nozzle. It is focused with a sheath flow of filtered air.
Particles in the focused air pass through the BARDet's chamber where they are interrogated by a
16mW CW laser beam generated by a diode laser operating at 375 nm wavelength (CUBE, Coherent).
The backward and forward scattered signals are detected with two PMTs (H6780, Hamamatsu)
mounted at the 35$^{\circ}$and 145$^{\circ}$ angles to the laser beam axis.

The fluorescence of particles is measured at a 90$^{\circ}$ angle to the laser beam with 32 channel PMT

(A10766, Hamamatsu). The longpass filter with cutting edge at 400 nm (Edmund Optics) separates
the fluorescence signal from scattered light. The multichannel PMT measures fluorescence in 18
active channels in a range of 415.4-643.5 nm. The channels are grouped in 7 bands. Such a solution
extends the dynamic range of measured spectra and, assures a high S/N ratio, and also reduces the
possibility of signal saturation. The remaining channels are not used. The band configuration is
presented in Table 1.

Table 1. Configuration of bands in the multichannel PMT.

| BARDet's Fluorescence Bands | Bandwidth [nm] |
|:---:|:---:|
| B1 | 415.4 – 429.3 |
| B2 | 443.1 – 456.8 |
| B3 | 470.5 – 484.2 |
| B4 | 497.8 – 524.9 |
| B5 | 538.3 – 565.0 |
| B6 | 578.3 – 604.6 |
| B7 | 617.6 – 643.5 |


**3.1.2.  Aerosols**

For the tests, dry powders of harmless substances were used since they did not need a

specialized aerosol protection chamber. In order to achieve a reliable aerosol classification, the ANNs
need to be trained possibly using a large number of measurement data. Therefore, various particle
types, that can be easily aerosolized, were tested.  Samples such as pollens, fungi, bacteria, spores
and plant debris naturally occur in the atmosphere. Biofluororphores such as riboflavin, cellulose,
amino acids and proteins were also characterized since they are present in biological materials. The
group of bacterial growth media was investigated due to their powerful influence on bacteria
fluorescence especially if they are not sufficiently washed. This can occur in the case of intentionally
released bacterial aerosols. Due to technical limitations, samples other than pharmaceutical could
not be aerosolized in this study. The aerosols of flours, and fluorescent non-biological substances
such as paper dust, AC fine Test Dust and talc were analyzed since they can occur especially in indoor
and public places. Non-fluorescent particles were not a subject of the research since they can be
automatically discarded as non-biologically applying given fluorescence thresholds.

The samples used for this study are listed in Table 2. To perform numerous experiments,

disposable vials were used, one for each aerosol sample. This prevented cross contamination
between measured samples. The aerosols were generated from modified 50 ml Falcon tubes placed

on the vortex. The vials in the lower part contained two connectors for silicon tubes. Vortexed
particles were entrained and formed an aerosol cloud inside the Falcon tube. The aerosolized
particles were aspirated from the vial to BARDet's aerosol inlet. Each tube contained about 50 mg of
the dry powder sample. During aerosol generation, filtered air was supplied into the vial to
compensate for the BARDet's flow. The concentration of the aerosols was adjusted with vibration
frequency of the vortex. The measurement started after the aerosol reached a homogeneous
concentration. The experimental setup is shown in Figure 1.

Table 2. List of all substances used in the experiment.

| | Abbreviation | Name | Size [µm] | AF | Source | Group |
|---|---|---|---|---|---|---|
| 1 | FM | Fluoromax green fluorescent 7 um microspheres | 6.25±0.91 | 0.92±0.02 | Thermo scientific | standard 1 |
| 2 | RIB | Riboflavin | 2.22±1.82 | 0.88±0.09 | Sigma-Aldrich | standard 2 |
| 3 | BGP | *Cynodon dactylon* (Bermuda grass) | 28.35±0.6 | 0.97±0.01 | Duke Sci. Corp. | pollens |
| 4 | CP | Zea mays (Corn) | 78.13±1.22 | 0.95±0.01 | Duke Sci. Corp. | |
| 5 | CA | *Corylus avellana* (Common hazel) | 27.71±1.33 | 0.67±0.04 | (*OC) | |
| 6 | LP | *Lycopodium* | 30.67±1.2 | 0.94±0.01 | Fluka | |
| 7 | PPP | *Poa pratrensis* (Kentucky bluegrass) | 30.62±0.87 | 0.94±0.01 | Sigma-Aldrich | |
| 8 | RP | *Ambrosia* (Ragweed) | 19.48±0.78 | 0.99±0.01 | Duke Sci. Corp. | |
| 9 | SCP | *Secale cereale* (Rye) | 44.8±2.01 | 0.94±0.01 | Sigma-Aldrich | |
| 10 | SP | *Picea* (Spruce) | 70.09±4.16 | 0.88±0.02 | (*OC) | |
| 11 | AA | *Abies alba* (Silver fir) | 84.56±12.77 | 0.92±0.02 | (*OC) | |
| 12 | UDP | *Urtica dioica* (Common nettle) | 14.99±1.26 | 0.9±0.05 | (*OC) | |
| 13 | PSP | *Pinus sylvestris* (Scots pine) | 39.29±1.44 | 0.93±0.02 | (*OC) | |
| 14 | PNP | *Pinus nigra* (Black pine) | 44.97±1.33 | 0.88±0.03 | (*OC) | |
| 15 | LPP | *Lycopodium* (Poland) | 28.66±0.6 | 0.95±0.01 | (*OC) | |
| 16 | PMP | *Broussonetia papyrifera* (Paper mulberry ) | 13.57±0.88 | 0.94±0.04 | Duke Sci. Corp. | |
| 17 | ATP | *Artemisia tridentata* (Big Sagebrush) | 22.53±0.42 | 0.96±0.01 | Sigma-Aldrich | |
| 18 | AAP | *Artemisia absynthium* (Wormwood) | 18.37±1.51 | 0.96±0.02 | Sigma-Aldrich | |
| 19 | CPP | *Chenopodium* | 27.29±0.97 | 0.98±0.01 | (*OC) | |
| 20 | BWF | Buck wheat flour | 25.17±15.76 | 0.82±0.06 | MELVIT Poland (*RS) | flours |
| 21 | PF | Potato flour | 21.23±3.11 | 0.96±0.03 | KUPIEC Poland (*RS) | |

| # | Abbr. | Name | | | Source | Category |
|---|---|---|---|---|---|---|
| 22 | RF | Rice flour | 18.22±6.23 | 0.6±0.07 | MELVIT Poland (*RS) | |
| 23 | TF | Tapioca flour | 12.91±3.41 | 0.7±0.06 | COCK BRAND (*RS) | |
| 24 | WF | Wheat flour | 20.57±4.36 | 0.62±0.07 | MELVIT Poland (*RS) | |
| 25 | Trp | Tryptophan | 15.42±8.96 | 0.81±0.08 | Sigma-Aldrich | amino acids and proteins |
| 26 | Phe | Phenylalanine | 10.41±5.31 | 0.73±0.11 | Sigma-Aldrich | |
| 27 | BSA | Bovine Serum Albumin | 63.8±30.49 | 0.43±0.05 | POCH Poland | |
| 28 | OVA | Ovalbumin | 26.45±5.31 | 0.83±0.07 | POCH Poland | |
| 29 | AMBAMB | *Bif. animalis, S. boulardii, S. thermophilus, L. casei, L. bulgaricus* | 27.97±4.42 | 0.84±0.03 | AMBIO Probiotyk, Lab. Galenowe Poland (*P) | bacteria in medium |
| 30 | LCB | *Lactobacillus bulgaricus* | 51.16±19.33 | 0.68±0.08 | LakciBios, ASA Poland (*P) | |
| 31 | LF | *Bifidobacterium animalis, L. acidophilus* | 32.62±8.45 | 0.82±0.07 | Linex forte, LEK Pharmaceuticals d.d. Slovenia (*P) | |
| 32 | BA | Bacteriological Agar | 49.47±10.03 | 0.74±0.07 | Sigma-Aldrich | medium |
| 33 | BAB | Blood Agar Base | 18.78±2.11 | 0.71±0.12 | Sigma-Aldrich | |
| 34 | LB | Luria broth | 15.11±6 | 0.67±0.07 | Sigma-Aldrich | |
| 35 | NB | Nutrient broth | 42.67±9.21 | 0.69±0.03 | Sigma-Aldrich | |
| 36 | BTSTG | *Bacillus thuringiensis* spores technical grade | 7.13±5.95 | 0.72±0.12 | Agricultural | Bacterial spore with admixtures |
| 37 | SB | *Saccharomyces boulardii* | 57.82±7.56 | 0.69±0.05 | Enterol, Biocodex France (*P) | fungi with admixtures |
| 38 | SC | *Saccharomyces cerevisiae* | 21.33±5.55 | 0.76±0.07 | Dr. Oetker Germany (*RS) | |
| 39 | LS | *Lycoperdon* spores | 14.52±0.62 | 0.92±0.02 | (*OC) | fungal spores |
| 40 | JGSS | Johnsons grass smut spores | 6.91±0.34 | 0.98±0.02 | Duke Sci. Corp. | smut spore (fungal spore) |
| 41 | BGSS | Bermuda grass smut spores | 6.47±0.27 | 0.97±0.02 | Duke Sci. Corp. | |
| 42 | ACFTD | AC Fine Test Dust | 3.47±2.34 | 0.87±0.09 | Duke Sci. Corp. | other |
| 43 | NT | Nivea talc | 14.33±4.71 | 0.77±0.09 | Nivea Baby (*RS) | |
| 44 | PPD | Printer paper dust | 76.37±18.89 | 0.43±0.11 | XEROX Laserprint collected from paper shredder (*RS) | |
| 45 | PTD | Paper towel dust | 73.45±25.65 | 0.56±0.15 | Merida Poland collected from crushed towel (*RS) | |
| 46 | CIN | Cinnamon | 23.97±4.39 | 0.78±0.05 | Kamis Poland (*RS) | |
| 47 | CEL | Celulose | 82.86±14.28 | 0.25±0.04 | Sigma-Aldrich | |

| 48 | GGL | Ground Green Leaves | 18.03±4.3 | 0.77±0.09 | Dried and ground Oak (*OC) | |

*OC – pollens collected from trees, flowers and grass at the region of Warsaw during vegetative seasons in 2015 and 2016.

*RS – Regular shops in Warsaw where common goods are purchased.

*P – Pharmacy shops in Warsaw

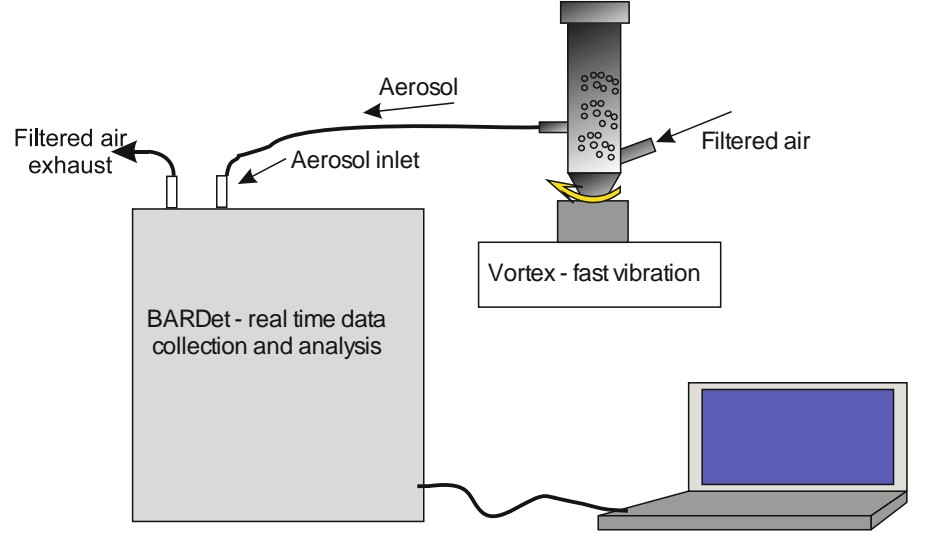

Figure 1.  Setup of aerosol generation, data recording and analysis.

### 3.1.3.  Aerosol microscopy

For microscopy analysis the aerosols were generated as described above and collected by impaction on a glass microscopic slide. The visualization of the samples was performed using a Nikon Eclipse Ti-U microscope with 10x objective. The images were recorded with a 5-megapixel DS-Fi1 camera. The aerosol equivalent diameters and circularity were analyzed automatically using NIS-Elements 64bit 3.22.10 software. The threshold of particle outline was corrected manually to obtain the visually best fit.

### 3.1.4.  Data acquisition method and pre-processing

The fluorescence of each particle was recorded in 7 bands. This creates a time series of the signals which has to be pre-processed before further analysis. There are two steps in gathering data. The first one is performed by the internal BARDet's software which is responsible for controlling the instrument and the acquisition of raw signals. Then data is forwarded to a pre-processing module in the analysis software. Its first task is to extract valuable signals from the noise (three sigma rule). After that a normalization procedure is required. It is performed first by subtracting the average value of the signal and then normalizing it to its standard deviation. The main goal was to analyze the shape of the emission spectrum (not signal strength). An example visualization of input data is shown in Figure 2.

The data acquisition process started after the stabilization of the aerosol generation rate which was measured by the device.  It was important not to exceed one particle per 2 ms of data integration

time in a 20 us measurement window. Finally, a total of 114,779 spectral characteristics of 48 aerosols was gathered, which gives on average 2391 (standard deviation 437) fluorescence characteristics per substance. From the recorded data 80% was used as a training data set and 20% as a test data set.

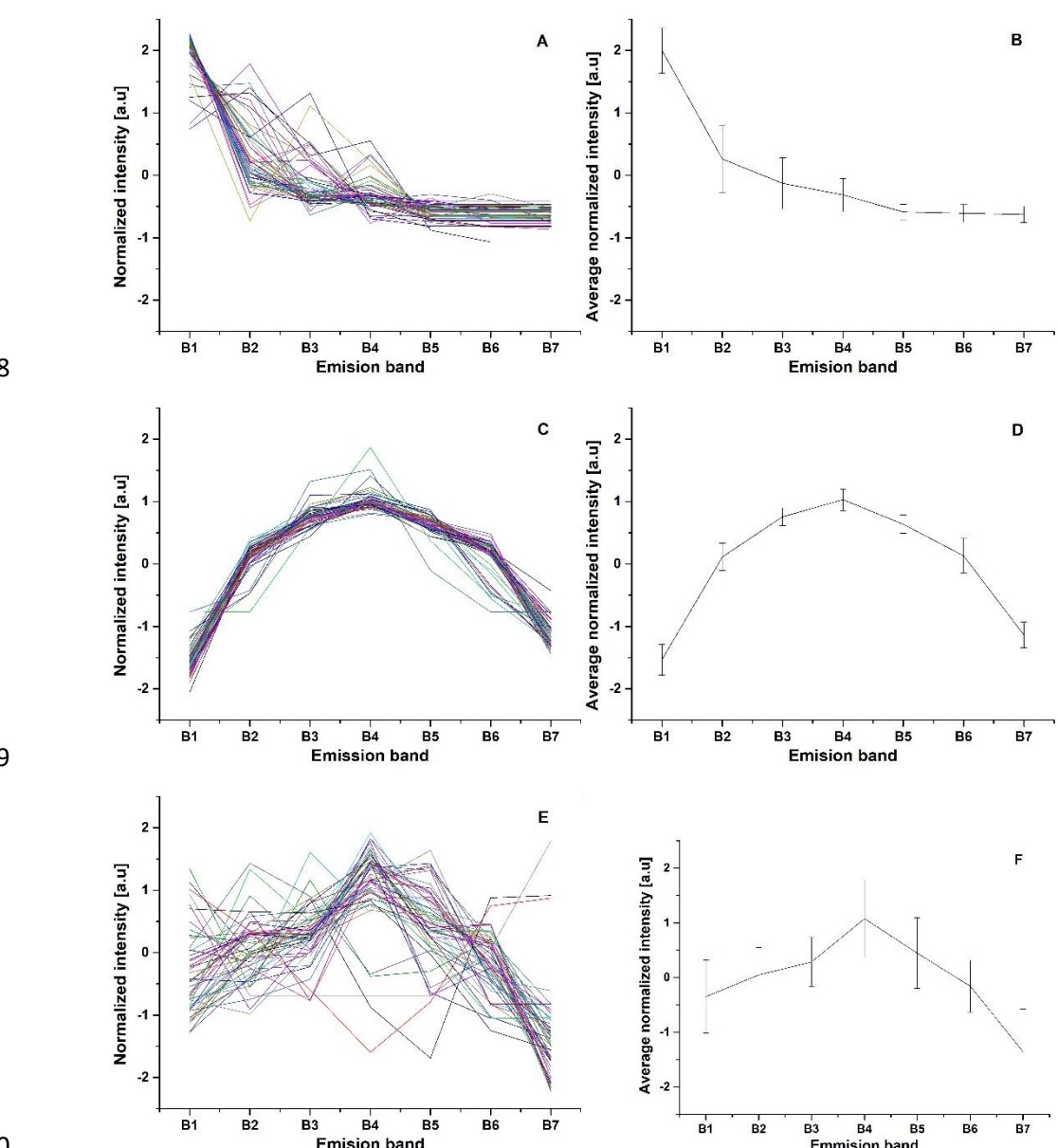

Figure 2.  Example, normalized 50 subsequent fluorescence characteristics of NT (A), FM (C) and LCB (E) and corresponding averaged normalized intensities of NT (B), FM (D) and LCB (F). Error bars represent standard deviation of measurements.

## 3.2.     Data analysis
### 3.2.1.  ANN (Artificial Neural Network)
#### 3.2.1.1.     Basics

There are many types of Artificial Neural Networks (ANNs), but in this paper only the

backpropagation algorithm is demonstrated because it is one of the most practical ones. The main
concept of this algorithm is based on a model of the neuron that has two tasks. It aggregates signals
(1) and then processes them by an activation function (2), which, in this research, is a sigmoid. The
result of such single processing is a new signal $z_j$ propagated to other neurons (Figure 3).

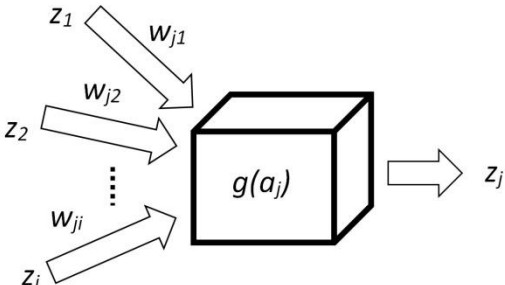


Figure 3.  Mathematical model of single neuron cell.

$$a_j = \sum_i w_{ji} z_i \tag{1}$$


$a_j$- aggregated signal, $w_{ji}$- weight that connects neuron *i* with *j*, $z_i$- signal (input).

$$g(a_j) = \frac{1}{1 + e^{-\beta a_j}} \tag{2}$$


$g(a_j)$ – sigmoidal function, $\beta$- parameter (steepness) of sigmoid curve.

The structure of a neural network is formed by layers of neurons: input, hidden and output. In

this research input neurons constitute a fluorescence spectrum and output neurons represent
substances. Most computations are carried out in the hidden layers (no more than two layers were
examined). The schematic representation of neuron layers is presented in Figure 4.

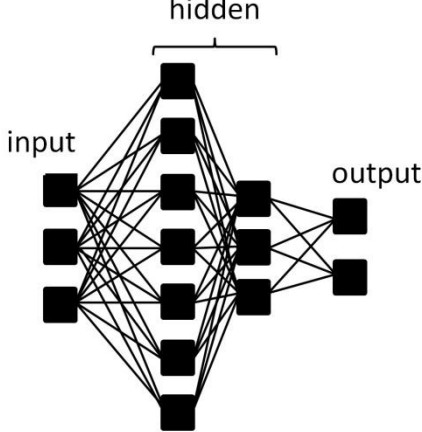


Figure 4. Typical topology of an artificial neural network.

The described algorithm constitutes the supervised learning method that requires training data

for a teaching process. This allows one to calculate an error between the target shown and the ANN
response. Every problem is related to minimizing output error which is calculated as Mean Squared
Error (3).

$$E = \frac{1}{2} \sum_{k=1}^{c} (y_k - t_k)^2 \tag{3}$$

$E$ – Mean Squared Error, $t_k$ - observed value (target), $y_k$ - calculated response, $k$-output neuron, $c$ –
number of output neurons.
The gradient descent method is used to find a minimum of error function. Error is dependent on
network weights $\Delta w_{ji}$ which might be adjusted (4). In order to update weights correctly, firstly one
needs to propagate error backwards by calculating partial derivatives $\delta_j$ (5) (Figure 5). All
mathematical details are well described by C. M. Bishop (Bishop, 1995).

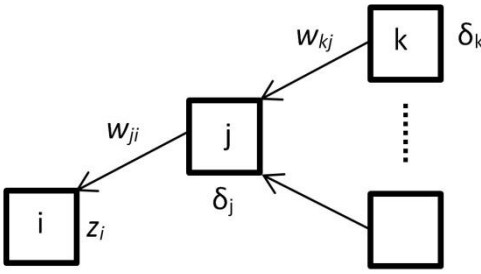


Figure 5. Model of backward error propagation.

$$\Delta w_{ji}(t) = -\eta \delta_j z_i + m \Delta w_{ji}(t-1) \tag{4}$$

$\eta$ - learning rate, $m$ - momentum, $t$ - iteration.

$$\frac{\delta E}{\delta w_{ji}} = \frac{\delta E}{\delta a_j} \frac{\delta a_j}{\delta w_{ji}} = \delta_j z_i \qquad\qquad \delta_j = g'(a_j) \sum_{k} w_{kj} \delta_k \tag{5}$$

The learning rate factor determines the size of the steps while the momentum parameter
enables the local minimum to be omitted by adding a fraction of the weight correction from the last
step.
After the correction of all weights of the ANN, the output error is examined, and the procedure
starts again unless an error level is low enough and there is no overfitting. All data are divided into
three different sets: training, test and validation. For calculations during the learning process, only
the first two are used. In order to determine whether it is time to stop the teaching process, one has
to observe an error in the test set. There will be a moment when this error comes to be constant or
starts increasing due to the overfitting of training data (Figure 6). The validation data set may be
useful for comparing different models or just to verify the current model on a completely separate
set of data.

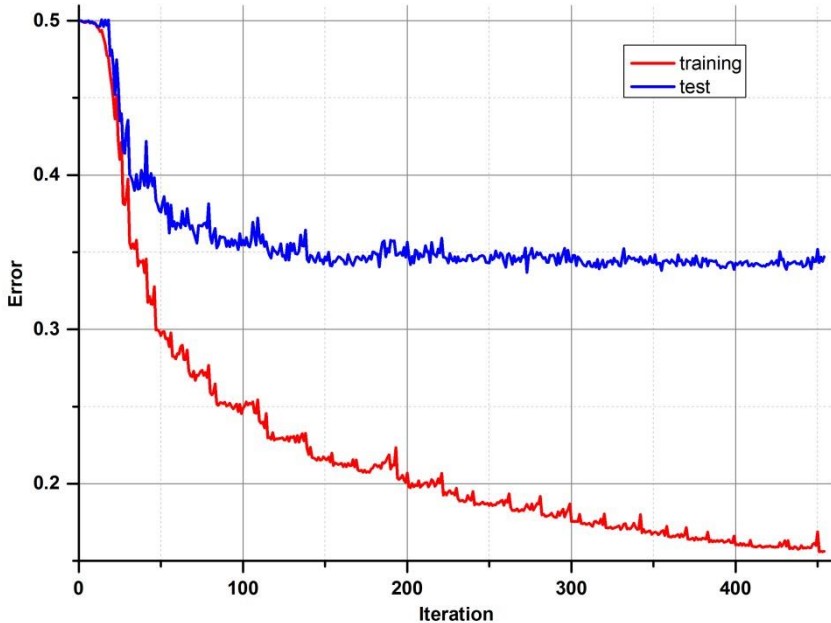

Figure 6. Example of error minimizing during the training process.
### 3.2.1.2.    Implementation of ANN for BARDet
There are statistical commercial software packages available that provide ANN modules as one
of the methods to analyze the data. It is worthwhile noting that customized software was developed
for this research. This approach helped us to understand ANNs in depth and led to the development
of software that is not only responsible for data pre-processing and network training, but also
(mainly) for solving a real time classification problem.
Ruske et al. in their studies (Ruske et al., 2017) compared various algorithms to analyze single
particle data and noted that an ANN requires much more user input. However, we present a method
to overcome this inconvenience by automating the process and implementing procedures which
simplify and improve the analysis.
The main disadvantage of an ANN is the fact that it is a parametrized algorithm. How well it
works depends strictly on a proper choice of the best possible factors, which may be different for
each problem. There are two types of factors that influence the ANN outcome. The first one
corresponds to the architecture of the ANN which comprises a number of layers, neurons and an
activation function parameter. The second one determines the learning process: momentum and
learning rate. The latter can be tuned during the learning process to make it much faster. The "bold
driver" procedure was chosen for that purpose. It continuously increases the learning rate unless an
error is higher from that before the change. If it is, the algorithm radically decreases the learning rate
and obtains weights from the last step again. Teaching an ANN is a stochastic process initiated by
using randomly chosen initial weights. It was found that the best procedure for this investigation
would be to conduct all optimization processes that way. Therefore, the parameters of the ANN,
responsible both for structure and learning process, are randomly selected until the desired result is
reached. In fact, the calculations are carried out automatically and simultaneously for several models
by means of multi core-oriented software. The benefits of this approach are time saving and high
levels of efficiency and effectiveness in finding the best model. The latter is especially important,
because the goal is to create a model that produces the best results, which doesn't necessary mean
creating a more complicated network (more neurons or layers).
### 3.2.2.  Model evaluation
The main goal of the analysis described in this paper is to find a solution to the bio-aerosol
classification problem. When a training process ends, a final model is created, a network, which has a
unique structure and a set of weights. One can create many of them and make a comparison only by
using the final error. It is not the best solution, because the goal is to distinguish patterns in data
consistently, not to produce a network with a minimal error. That is why there is a need to make a
final analysis of the results and evaluate the model in accordance with the best classification
performance.
The standard method for visualization of results is a confusion matrix which will be necessary for
Receiver Operating Characteristics (ROC) analysis (Fawcett, 2006). It simply shows what fraction of
population for each class is predicted correctly or not. Each element from the data set is assigned to
one of the following fits of the confusion matrix: True Positive (TP), True Negative (TN), False
Negative (FN) and False Positive (FP). If it belongs to TP and TN, it was classified correctly.
The ROC graphs are very simple but useful tools for discovering whether a classifier is worth
using or if it makes a random classification. It is based on two rates from the confusion matrix: hit
rate (6) and false alarm rate (7).

$$hit\ rate\ (true\ positive\ rate)$$
$$= \frac{TP}{TP + FN} \tag{6}$$


$$false\ alarm\ rate\ (false\ positive\ rate)$$
$$= \frac{FP}{FP + TN} \tag{7}$$

Each discrete classifier has a threshold level that assigns an element to a positive or negative
class. The points on the ROC graph (Figure 7) represent the classifier for many thresholds. The most
desirable curve will be obtained when the true positive rate is high, and the false positive rate is low
(convex line).  The random classifier, in turn, has a hit rate equal to a false alarm rate despite
threshold variation (diagonal line). To identify an ROC analysis with one coefficient, the area under
the curve (AUC) may be used. The higher value of AUC results in better performance (0.5 means
random, 1 - excellent).

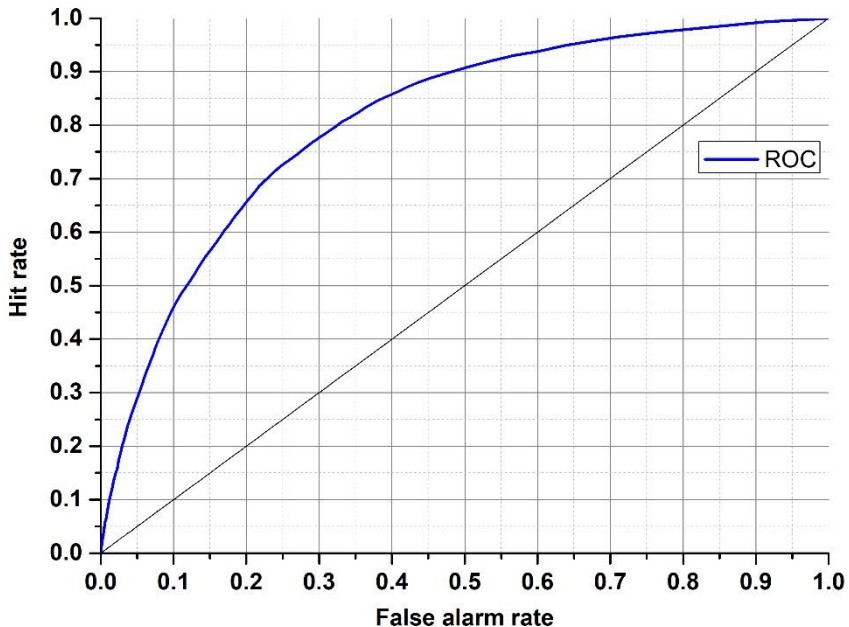


Figure 7. ROC graph with an example of classifier (blue).
The confusion matrix and ROC analysis described above were defined for two class problems
(positive, negative). There is a straightforward way to expand it for multi-class problems. One needs
to take a desired class versus all other classes. Then it will be possible to compare how good the
classifier for specific classes within one model is.
4. **Results**
**4.2. ANN performance**
The first attempts were made to distinguish all substances using only one neural network model.
The tests revealed that it is impossible due to the huge number of samples (48 aerosols) and only a
few of them presented significantly different fluorescence spectra which allow accurate
characterization. The remaining substances are then misclassified. Therefore, we decided to use a
more practical approach to this problem, which would be to create several groups (considering
information about aerosols), but we did not want to make any classes *a priori*. Although the ANN
type demonstrated needs training, which requires a set of known classes, further tests showed that
there is a possibility of finding similarities between substances through the analysis of confusion
matrices. It was achieved after many trials of matching substances, which were not well separated,
into new groups and checking if they are good enough on ROC graphs. Consequently, this procedure
was also applied to those new groups.

All examples demonstrated below were calculated on the test data sets, not training data. In the
first presented (Figure 8), which tries to classify all of the 48 substances (group 0), four aerosols
reached a very high accuracy of separation (AUC>0,9). The best separation was achieved for
fluorescent microspheres (FM). In this case 98.5% of all FM particles were correctly classified.
Similarly, an efficient separation was achieved for riboflavin (RIB), Talc (NT) and *Lactobacillus*
*bulgaricus* (LCB). The remaining aerosols were divided into 3 separate groups that gather the most
similar substances (group 1-3) (Table 3). The subsequent groups up to 21 represent individual ANNs
leading to the final classification of the aerosol. In practice separation is done not by one confusion
matrix (ANN) but by all of them in sequence (22 ANNs combined in a decision tree). For example, if
an ANN classifies unknown substance into any of 22 groups it means that decision process is not
ended but from that moment another ANN classifies this substance. However, each new ANN is
trained using only a subsection of the data excluding the data from other groups.
Table 3. Exemplary confusion matrix of all aerosols classified by the first ANN.

| | | predicted | | | | | | |
|---|---|---|---|---|---|---|---|---|
| | | FM | RIB | NT | LCB | group 3 | group 1 | group 2 |
| **true** | FM | 98.5 | 0 | 0 | 0.3 | 0.1 | 0 | 1.1 |
| | RIB | 0.1 | 91 | 0.5 | 3.1 | 1.2 | 0.6 | 3.4 |
| | NT | 0 | 0.1 | 86.5 | 0 | 9.3 | 0.3 | 3.8 |
| | LCB | 1 | 1.6 | 0.6 | 72.7 | 3.9 | 10.7 | 9.5 |
| | group 3 | 0 | 0.7 | 6.6 | 0.6 | 63.3 | 12 | 16.8 |
| | group 1 | 0.2 | 1 | 1 | 7.9 | 12.5 | 61.6 | 15.8 |
| | group 2 | 0.1 | 1.2 | 3.8 | 6.6 | 17.6 | 13.2 | 57.4 |

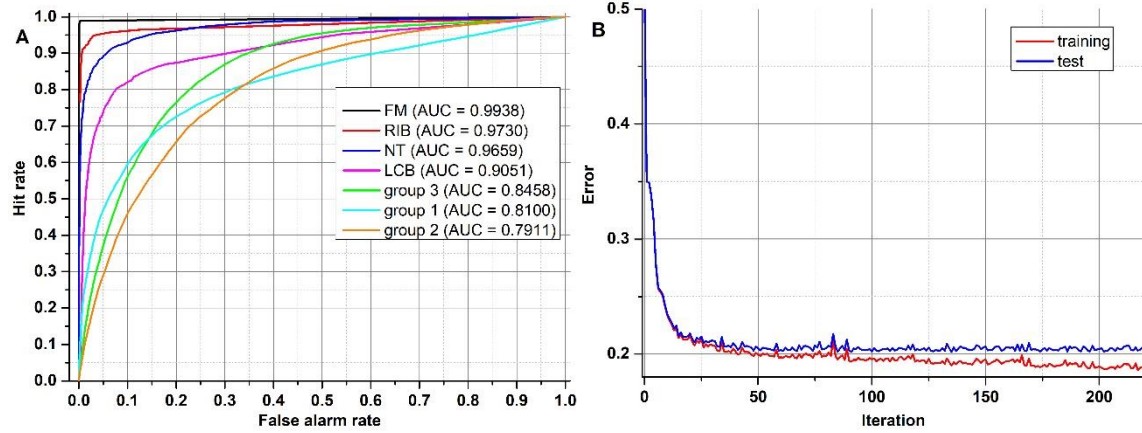


Figure 8. (A) ROC and (B) error progress of ANN that classifies all samples.
Table 4 and Figure 9 show results achieved for two substances that have a very similar spectrum
and the AUCs calculated are not much higher than in a random classifier. This example clearly shows
why we are not always able to classify every single particle of aerosol with 100% accuracy. However,
just a representative number (several dozen) of measured particles (a cloud) allows the proper
prediction of aerosol types within a few seconds. This is easy to observe during real time detection,
because counts allocated in a confusion matrix tend to reach a stable state quite quickly.

| | | predicted | |
|---|---|---|---|
| | | BWF | CEL |
| **true** | BWF | 54.8 | 45.2 |
| | CEL | 45.6 | 54.4 |

Table 4. Confusion matrix of two substances that have very similar spectra.

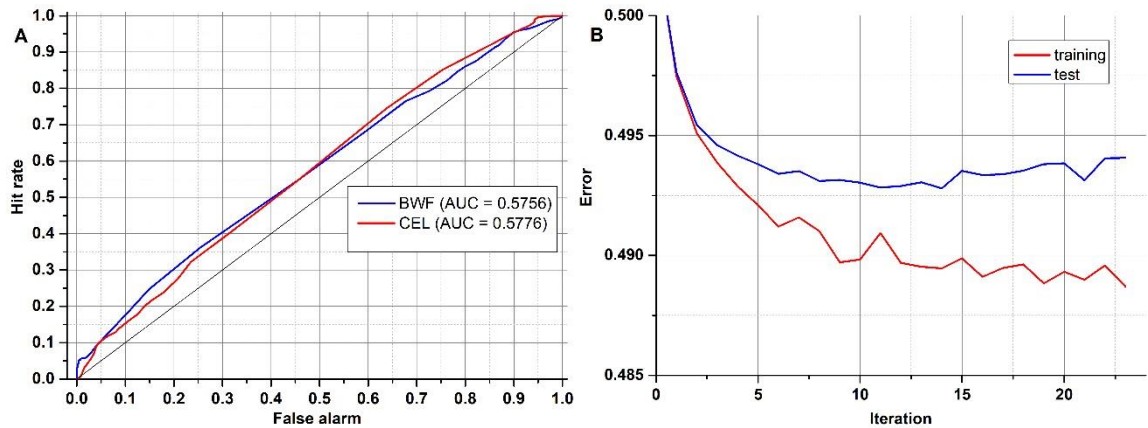


Figure 9. ROC (A) and error progress (B) of ANN which classify two very similar samples.


**4.3. Classification tree**

Finally, to achieve the best possible classification, a decision tree was created (Figure 10). It
comprises not one, but 22 models. The process of creating them is not replicable in terms of the
exact factors used for ANN generation. However, this is not essential, because the decision tree is
based on ANN results (classification ability), which should be possibly the highest. Therefore, the final
result will be the same. It is difficult to present confusion matrices and ROC graphs for all neural
networks in this paper. Therefore, only the most interesting one has been discussed. Here, each node
represents a network that classifies a group of aerosols. The aerosols on the left side of the diagram
show the most distinct differences, thus they are easy to classify (Level 0). On the right side (Level 1-
5), this task is much more demanding due to a similar spectrum and the separation is less probable in
accordance with single particles, although it is still very useful from a practical point of view for
aerosol cloud discrimination.

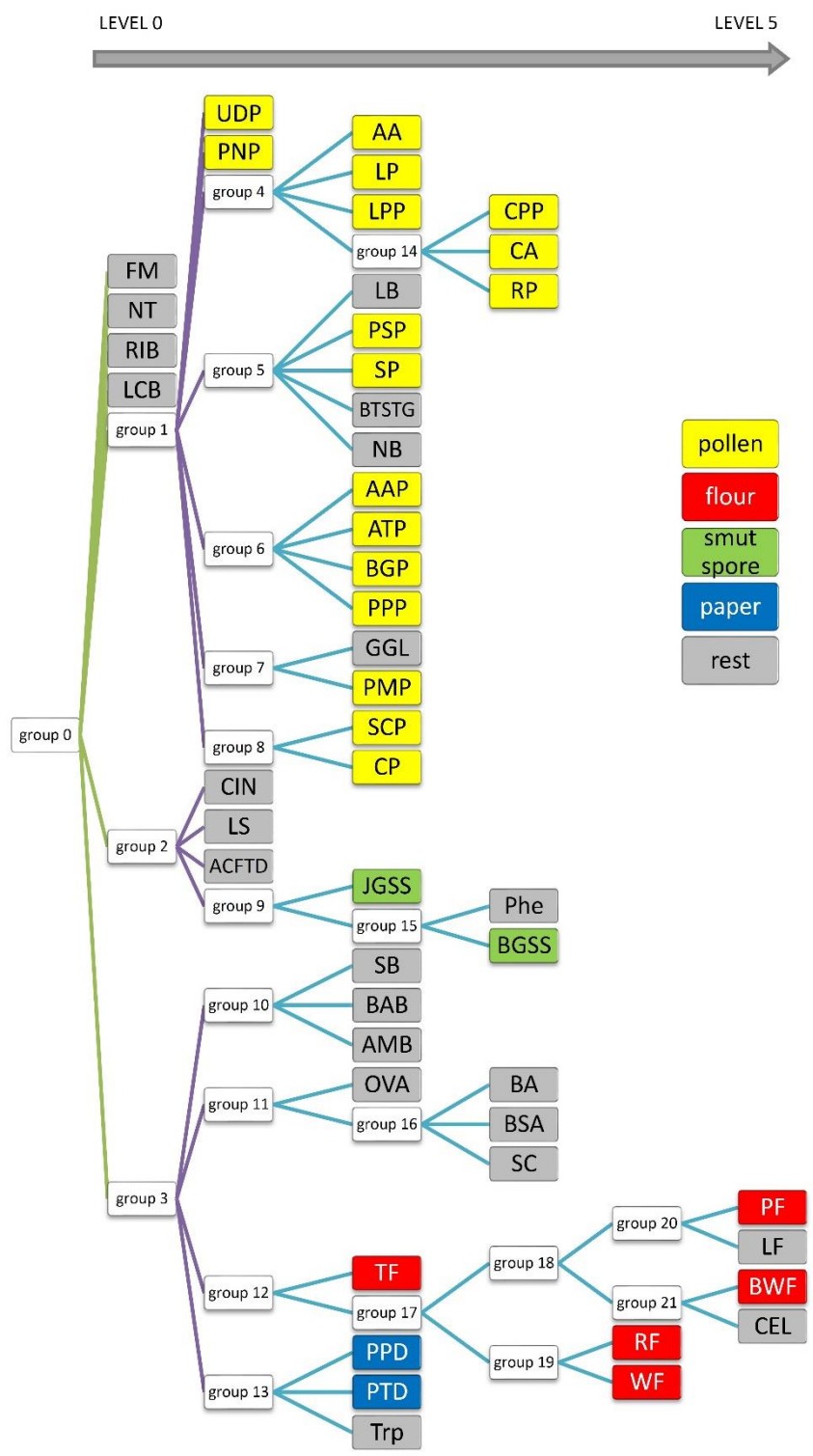

Figure 10. The decision tree consists of 22 ANNs separating 48 substances.
At first glance one can see that FM and RIB are very well recognized, but that was expected
because these are standards of fluorescence. Surprisingly, NT and LCB aerosols were also separated
from the others (level 0 network). Further analysis of the tree structure identifies a correlation
between samples and their real categories. It is especially noticeable for pollens, which are allocated
to a separate branch of that tree, and all stems from group 1. Most of them were classified on the
third level. Interestingly all grass pollens (AAP, ATP, BGP, PPP) belong to the same group, 6. Similarly,
both *Lycopodium* pollens from different regions of the word show a close correlation, although *Abies*
*alba*, which is a tree, was classified in the same group. Flours, Smut Spores and Papers are dispersed
between different levels, but particular groups belong to the same branch of the tree. However, some
of the samples are scattered on the whole tree area and do not correspond to any group.
It should be noted that the result is a system of 22 ANNs that work simultaneously. In
comparison to the training process, which is rather time consuming and has to be empirically
optimized, this cluster of learned ANNs delivers high performance. Input data is processed by a single
ANN in milliseconds. This performance makes the neural network a great tool as a splitting node in
the classification tree. Compared to our previous results, where Principal Component Analysis was
applied to analyze data from BARDet (Kaliszewski et al., 2016), the ANNs allowed much better
discrimination between various bio-aerosols.
5.  **Summary**
In this paper the possibility of applying an Artificial Neural Network (ANN) for real time
classification of biological aerosols was investigated. The spectral characteristics of bio-aerosols were
collected using the BARDet instrument. The database consisted of 48 substances. Finally, 22 neural
networks were trained and combined into a decision tree. This allowed aerosols to be
characterizedin real time. Tests revealed that only certain substances have such characteristic
fluorescence spectra that allow correct classification of almost each particle. However, in all other
cases the system was able to recognize a particular aerosol accurately with no mistake, but a
representative number of several dozens of particles in a cloud was necessary. Further
approximation was based on decision tree analysis where each node corresponded to a separate
learned ANN. The best sets of ANNs for each group of similar aerosols were discovered utilizing
confusion matrices and ROC analysis. Our intention was to make a complete system which detects
and classifies substances without creating groups *a priori*. This attitude helped us to create a
powerful analytical tool that works automatically, and the results of classification are immediately
available on the operator's screen.
This study proved that it is possible to create a tool for a highly effective analysis of bio-aerosols
using multiple ANNs combined into a decision tree. Our approach allowed us to automate and speed
up the analysis, which reduced time and the amount of computing power needed. In a future study
the database will be extended to obtain potentially a vast variety of samples including
atmospherically relevant bacteria and fungi. In the next step, the actual performance of the system
will be determined under real environmental conditions, which will be most challenging due to the
presence of unknown fluorescent and non-fluorescent particles.

**Data availability**
he experimental aerosol data can be provided upon request. The software for automatic data
analysis cannot be publicly provided at this moment since it is a subject of negotiations with a
company.

**Acknowledgments**
The work ppresented was supported by a grant from The National Centre of Research and
Development (Poland), within the project "Mobile laboratory for environmental sampling and
identification of biological threats" (O ROB 0031 01/ID/31/1).

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
