# Peer review of "Improved real-time bio-aerosol classification using Artificial Neural Networks"

_Atmospheric Measurement Techniques, 2018_

## Referee Comment (RC1) · Anonymous Referee #1 · 5 Jun 2018

This manuscript details the use of an Artificial Neural Network, or ANN, to attempt to better identify bio-aerosol. Bio-aerosol has been a topic of contemporary interest in the atmospheric sciences and neural networks have gained prominence as a data reduction and analysis technique. This is therefore a paper that could be of interest to the AMT readership. There are however several large missing sections, e.g. aerosol justification and characterization, that should be addressed before it is publishable.

1. The writing of the paper is a bit too familiar and there are many unquantifiable terms, e.g. "Society is awaiting anxiously for system that could inform them in real-time about a real danger that is suspended in the air." – this would be a rather improved paper if this type of writing could be toned down as in "There is a need for real-time information about ambient particulate matter." 2. In addition, the paper could benefit from a

through read from a native English speaker with a focus on removal of incorrect and non-scientific terms. Examples, but by no means comprehensive: "really promising", "very high performance", past tense of grind is ground, not grinded, etc. 3. The name of the technique to which the ANN is applied, BARDet, should be stated in the abstract. 4. The central issue with this paper is there needs to be a description of the aerosol generation method and the produced size distribution of each sample; some are solids, some are liquids. Were sizes comparable? Concentrations? Ideally this is a sub-section of 3.1.2. Going farther, why were these samples chosen? Some seem rather important e.g. pollens, while others are unclear. Paper towel? Multiple broths? It is upon the authors not to simply present so may aerosol types but instead (1) carefully and completely characterize the aerosol investigated – not only what they look like to the BARDet - and (2) to argue why they are being investigated (do they have any atmospheric importance which is the theme of the paper)? 5. Going a step further, although there are 48 aerosol types suggested, in practive the confusion matrix says the separation is based on 7 broader classes. If this is indeed the case (as it appears) then (1) the abstract should reflect separation of 7 classes, not the 48 stated (2) Table 1 should state what fits into each class, since this is the central concept 6. The statistic in Table 4 need to be placed in the abstract and repeated in the summary, these are the central results. For example, in Tables 4 and 5 it appears that there can be confusion on the 50th centile level. This is not altogether great separation and should be explicitly stated for the reader from the outset. The 48 types and 114k number of spectra, which are the data set, belong only in the methods section; while these seem rather impressive they are not results. The authors should therefore replace the sentences which repeat these values in abstract and summary with the separation ability. 7. Table 3 is overly simplistic for a table; this can be stated in a single sentence. Please remove. 8. In the summary : "This study proved that it is possible to create a tool for a highly effective analysis of bio-aerosols using multiple ANNs combined into decision tree." – this is again an unquantified statement. It is also at odds with "Tests revealed that only several substances have such characteristic fluorescence spectra that allows correct

classification of almost each particle. However, in all other cases the system was able to recognize a particular aerosol cloud." Please provide the separation ability and then let the reader judge is this is a highly effective analysis. 9. Why weren't non-biological materials tested?

―――――――――――――――

---

## Referee Comment (RC2) · Anonymous Referee #3 · 10 Jun 2018

In this paper the authors present a method for bio-aerosol classification using labelled laboratory data. The authors are correct in noting the need to improve and document such methods for improved bio-aerosol research. However before publication is considered, I feel the following points should be addressed. Presently it is unclear how anyone might replicate these results.

Minor points:

The formatting of references is wrong? Please check with the Copernicus guidelines and change from (xx)(xx) format to (xx;xx;xx...)

There is a range of grammatical issues that need revising before publication. I have listed some below but would suggest the authors re-read the paper and change ac-

cordingly, removing any vague descriptions that require support with numerics or information to enable replication of experimental conditions. E.g:  c Line 76: 'This paper focuses on the application of ANN for real time discrimination of bio-aerosols basing on single particle fluorescence characteristics.' Please change 'basing' to 'based'  c Line 108: 'The concentration of the aerosols was adjusted with vibration frequency of [the] vortex. '  c Line 176: In order to determine whether it is time to stop teaching,'. This is too informal. I would suggest re-writing in terms of the fitting process.

Specific Points:

In table 2 the authors use the term 'own collection'. I'm a little concerned this does not provide enough information to enable replication of results. Where was the sample obtained? How old? Also the terms 'regular shop' and 'pharmacy' raise similar concerns. Which Pharmaceutical brand?

Would it be possible to present size and shape information for each specie in a separate table?

Line 119: Please list the bands of florescence recorded. You have done so in Table 1 but you should reference this table in the text on this line to avoid confusion.

Line 127: 'An Important aspect of the data acquisition process was monitoring the rate of generation of aerosol, which should be stable (not too high or spontaneous). ' Please define how this is quantified. What is 'too high'? How would this experiment be repeated?

Line 130: 'It is important to note that fact because of its statistical value for the further analysis'. What statistical value?

Section 3.2.1.2: What comparisons have been made, if any, between the bespoke implementation of the ANN in this work with what should be identical performance in existing software packages? How do we know the implementation of the bespoke ANN is correct? Please provide evidence.

[Figure]

Major points:

It is difficult to contextualise the input data being used. Please provide a visualisation of some example spectra.

To the best of the reviewers understanding, each particle will be classified at multiple levels of the decision tree. For example each particle will be classified as FM7, Rib, NT, LCB, or group 1 etc. and then should the particle be identified as group 1, the particle will then get classified again as UDP, PNP, group 4 etc. For example, should a particle from group 2 be misclassified and placed into group 1, which will happen about 12% of the time, how does this error propagate down the tree? Will it be evenly distributed amongst UDP, PNP, group 4 etc. or will it be heavily weighted towards one class?

With the exception of the level 0 ANN, I assume that each of the ANNs are trained only on a subsection of the data. This needs to be clarified. For example the ANN for group 1, is trained in absence of the data from group 2 etc.

On line 245 it is stated that it is impossible to produce a single neural network to perform classification of all 48 classes. Need to be clear whether this means that it is impossible because of the number of classes, or that it is possible to create a single neural network but the classification error is unreasonably high. Would it be possible to produce a contour confusion matrix plot for the full 48 classes, for a single ANN and for the approach suggested in the manuscript, or to provide adjusted rand score or percentage of particles correctly classified to demonstrate whether better classification can be attained using the tree of ANNs as opposed to a single ANN?

How was the decision tree created? I.e. how it was decided which individual classes would be placed into group 1 through 3?

The authors have indicated on line 203 that the hyper-parameters of the ANNs have been randomly selected until the desired/best result is reached. In terms of reproducibility, it would be helpful to specify the range of parameters which were tested and

which of these options produced the best results. Also did each of the 22 networks utilise the same hyper parameters, or was this optimisation conducted for each of the 22 networks?

There is no discussion on either data or software availability. The authors need to consider the default Copernicus publishing rules and provide text that would allow others to request access to both the data and software. If this is restricted, it should be stated with the reasons why. https://www.atmospheric-measurement-techniques.net/about/data_policy.html

---

## Author Comment (AC1) · 13 Jul 2018

We would like to thank the Reviewer for evaluation of our manuscript. The detailed answers to the questions are as follows:

Reviewer #1

This manuscript details the use of an Artificial Neural Network, or ANN, to attempt to better identify bio-aerosol. Bio-aerosol has been a topic of contemporary interest in the atmospheric sciences and neural networks have gained prominence as a data reduction and analysis technique. This is therefore a paper that could be of interest to the AMT readership. There are however several large missing sections, e.g. aerosol justification and characterization, that should be addressed before it is publishable.
1. The writing of the paper is a bit too familiar and there are many unquantifiable terms, e.g. "Society is awaiting anxiously for system that could inform them in real-time about a real danger that is suspended in the air." – this would be a rather improved paper if this type of writing could be toned down as in "There is a need for real-time information about ambient particulate matter."

- The sentence was corrected.

2. In addition, the paper could benefit from a through read from a native English speaker with a focus on removal of incorrect and non-scientific terms. Examples, but by no means comprehensive: "really promising", "very high performance", past tense of grind is ground, not grinded, etc.

- The language correction was performed.

3. The name of the technique to which the ANN is applied, BARDet, should be stated in the abstract.

- The name of the device was added to the abstract.

4. The central issue with this paper is there needs to be a description of the aerosol generation method and the produced size distribution of each sample; some are solids, some are liquids. Were sizes comparable? Concentrations? Ideally this is a sub-section of 3.1.2.

- All aerosols were generated from powders only as it was described in section 3.1.2. The sizes depend on dimensions of particles. An information about particle's sizes was added to the table 2.

Going farther, why were these samples chosen? Some seem rather important e.g. pollens, while others are unclear. Paper towel? Multiple broths? It is upon the authors not to simply present so may aerosol types but instead (1) care-fully and completely characterize the aerosol investigated – not only what they look like to the BARDet - and (2) to argue why they are being investigated (do they have any atmospheric importance

which is the theme of the paper)?

- The following explanation was added to the 3.1.2 section: "In order to achieve reliable aerosol classification the ANN's needs to be trained using possibly large number of measurement data. Therefore, various particle types, that can be easily aerosolized, were tested. Samples like pollens, fungi, bacteria, spores and leaves scraps are present in the atmosphere. Biofluorophores like riboflavin, cellulose, aminoacids and proteins were also characterized since they are components of biological materials. The group of bacterial growth media was investigated due to their strong influence on bacteria fluorescence especially if they are not sufficiently washed. This can occur in case of intentionally released bacterial aerosols. Due to technical limitations the other than pharmaceutical samples could be aerosolized in this study. The aerosols of flours, and fluorescent non-biological substances like paper dust, AC fine Test Dust and talc were analyzed since they can occur especially in indoor and public places. The non-fluorescent particles were not a subject of the research since they can be automatically discarded as non-biological applying given fluorescence threshold."

5. Going a step further, although there are 48 aerosol types suggested, in practive the confusion matrix says the separation is based on 7 broader classes. If this is indeed the case (as it appears) then (1) the abstract should reflect separation of 7 classes, not the 48 stated (2) Table 1 should state what fits into each class, since this is the central concept.

- In the manuscript we have stated as follows: "It is difficult to present confusion matrices and ROC graphs for all neural networks in this paper, so only the most interesting one has been discussed." - In practice separation is done not by one confusion matrix (ANN) but by all of them in sequence (22 ANN's combined in a decision tree). For example, if ANN classifies unknown substance into any of 22 groups it means that decision process is not ended but from that moment another ANN classifies this substance. That's why there are substances which only needs one ANN to make a classification (e.g. FM7), but there are also such which needs 6 ANN (e.g. BWF) to complete the

task. The main difference between this two examples is that 98.5% of all FM7 particles are classified correctly, but BWF has only 54.8% detected particles. However in both cases system recognize aerosol type every time with no mistake.

6. The statistic in Table 4 need to be placed in the abstract and repeated in the summary, these are the central results.

- Table 3, previously Table 4 do not represents the central result. It is only 1 of 22 nodes of a decision tree. The most important fact is that each one aerosol type can be recognized. In the abstract we added as follows: "In both cases the system recognized aerosol type with no mistake."

For example, in Tables 4 and 5 it appears that there can be confusion on the 50th centile level. This is not altogether great separation and should be explicitly stated for the reader from the outset.

- It was stated in the text. However, we hope that modified explanation will be helpful (Lines 451-456).

The 48 types and 114k number of spectra, which are the data set, belong only in the methods section; while these seem rather impressive they are not results. The authors should therefore replace the sentences which repeat these values in abstract and summary with the separation ability.

- We are agree with reviewer that number of data are not a result. Therefore they were removed from the abstract and summary.

7. Table 3 is overly simplistic for a table; this can be stated in a single sentence. Please remove.

- The sentence was added and the table was removed (Lines 382 – 384).

8. In the summary : "This study proved that it is possible to create a tool for a highly effective analysis of bio-aerosols using multiple ANNs combined into decision tree." –

this is again an unquantified statement. It is also at odds with "Tests revealed that only several substances have such characteristic fluorescence spectra that allows correct classification of almost each particle. However, in all other cases the system was able to recognize a particular aerosol cloud." Please provide the separation ability and then let the reader judge is this is a highly effective analysis.

- We provided for the reader only two examples that shows good and poor separation in accordance for individual particle within only these two groups (group 0 and group 21). Probably it was not emphasized clearly enough in the manuscript that system recognize aerosol type (all of them) with no mistake every time and that was main goal to achieve in presented analysis. - In the lines 581-583 we added as follows: "However, in all other cases the system was able to recognize a particular aerosol accurately with no mistake, but a representative number of several dozens of particles in a cloud was necessary."

9. Why weren't non-biological materials tested?

- The materials and methods section was improved. We justified the use of tested samples. We also changed confusing title in 3.1.2 "Bioaerosols" for "Aerosols" - The non-biological materials were tested: Fluoromax microspheres 7 um Nivea talc Printer paper dust Paper towel dust AC Fine test dust (This one can contain also biological particles) - The most of non-biological materials like gypsum, syloid, desert sand are non-fluorescent and there is no any problem to differentiate them from biological particles.

---

## Author Comment (AC2) · 13 Jul 2018

We would like to thank the Reviewer for evaluation of our manuscript. The detailed answers to the questions are as follows:

Reviewer #3

In this paper the authors present a method for bio-aerosol classification using labelled-laboratory data. The authors are correct in noting the need to improve and document such methods for improved bio-aerosol research. However before publication is considered, I feel the following points should be addressed. Presently it is unclear how anyone might replicate these results.

Minor points: The formatting of references is wrong? Please check with the Copernicus

guidelines and change from (xx)(xx) format to (xx;xx;xx...)

- The formatting of references was corrected.

There is a range of grammatical issues that need revising before publication. I have listed some below but would suggest the authors re-read the paper and change accordingly, removing any vague descriptions that require support with numerics or information to enable replication of experimental conditions. E.g: Line 76: 'This paper focuses on the application of ANN for real time discrimination of bio-aerosols basing on single particle fluorescence characteristics.' Please change 'basing' to 'based'

- Corrected

Line 108: 'The concentration of the aerosols was adjusted with vibration frequency of [the] vortex.

- Corrected

Line 176: In order to determine whether it is time to stop teaching,. This is too informal. I would suggest rewriting in terms of the fitting process.

- In our opinion "teaching" process is appropriately used phrase and is widely applied in ANN related literature. We used "overfitting" in context of data not the learning process.

Specific Points: In table 2 the authors use the term 'own collection'. I'm a little concerned this does not provide enough information to enable replication of results. Where was the sample obtained? How old? Also the terms 'regular shop' and 'pharmacy' raise similar concerns. Which Pharmaceutical brand?

- The description and full information on the samples was added to the table 2.

Would it be possible to present size and shape information for each specie in a separate table?

- The missing data were added to the table 2.

Line 119: Please list the bands of florescence recorded. You have done so in Table 1 but you should reference this table in the text on this line to avoid confusion.

- The table has been referenced in the text just above.

Line 127: 'An Important aspect of the data acquisition process was monitoring the rate of generation of aerosol, which should be stable (not too high or spontaneous). 'Please define how this is quantified. What is 'too high'? How would this experiment be repeated?

- The BARDet's measurement window is 20us, but the data are integrated and recorded every 2 ms. It gives up to 100 averaged aerosol characteristics per 2 ms. It does not strongly influence the result if one aerosol type is measured, however, we tried to avoid such measurements. - The sentence in the manuscript was clarified (Lines: 335-337) as follows: "The data acquisition process started after stabilization of aerosol generation rate which was measured by the device. It was important to not exceed one particle per 2 ms of data integration time at 20 us measurement window."

Line 130: 'It is important to note that fact because of its statistical value for the further analysis'. What statistical value?

- The sentence was removed.

Section 3.2.1.2: What comparisons have been made, if any, between the bespoke implementation of the ANN in this work with what should be identical performance in existing software packages? How do we know the implementation of the bespoke ANN is correct? Please provide evidence.

- The presented ANNs were not compared to existing packages. We believe that our implementation of ANNs is correct since they produce correct results on approximated mathematical functions.

Major points: It is difficult to contextualise the input data being used. Please provide a visualization of some example spectra.

- An exemplary characteristics were added as a figure 2.

To the best of the reviewers understanding, each particle will be classified at multiple levels of the decision tree. For example each particle will be classified as FM7, Rib, NT, LCB, or group 1 etc. and then should the particle be identified as group 1, the particle will then get classified again as UDP, PNP, group 4 etc.

- Yes. In practice separation is done not by one confusion matrix (ANN) but by all of them in sequence (22 ANN's combined in a decision tree). For example, if ANN classifies unknown substance into any of 22 groups it means that decision process is not ended but from that moment another ANN classifies this substance. That's why there are substances which only needs one ANN to make a classification (e.g. FM7), but there are also such which needs 6 ANN (e.g. BWF) to do that. Main difference between this two examples is that 98.5% of all FM7 particles are classified correctly but BWF has only 54.8% detected particles. However in both cases system recognize aerosol type every time with no mistake.

For example, should a particle from group 2 be misclassified and placed into group 1, which will happen about 12% of the time, how does this error propagate down the tree? Will it be evenly distributed amongst UDP, PNP, group 4 etc. or will it be heavily weighted towards one class?

- Error should be distributed according to confusion matrix of the group where particle is classified. There are 22 groups/ANN's/confusion matrices. In paper only 2 were presented as an examples.

With the exception of the level 0 ANN, I assume that each of the ANNs are trained only on a subsection of the data. This needs to be clarified. For example the ANN for group 1, is trained in absence of the data from group 2 etc.

- It is done exactly like that. - To clarify the text to the reader the following sentence in lines 504-516 was added: "In practice separation is done not by one confusion matrix

(ANN) but by all of them in sequence (22 ANN's combined in a decision tree). For example, if ANN classifies unknown substance into any of 22 groups it means that decision process is not ended but from that moment another ANN classifies this substance. However, each new ANN is trained using only subsection of the data excluding the data from other groups."

On line 245 it is stated that it is impossible to produce a single neural network to perform classification of all 48 classes. Need to be clear whether this means that it is impossible because of the number of classes, or that it is possible to create a single neural network but the classification error is unreasonably high.

- Our intention was to reporting that it is impossible to distinguish all substances using one ANN, not to create such single ANN. - In the manuscript it was as follows: "First attempts were made to distinguish all substances using only one neural network model. The tests revealed that it is impossible due to the huge number of samples (48 aerosols) and only a few of them presented significantly different fluorescence spectra." - To clarify the text in lines 487-488 where additional explanation was added: "…that allow accurate characterization. The remaining substances are then misclassified. Therefore, we decided to use a…."

Would it be possible to produce a contour confusion matrix plot for the full 48 classes, for a single ANN and for the approach suggested in the manuscript, or to provide adjusted rand score or percentage of particles correctly classified to demonstrate whether better classification can be attained using the tree of ANNs as opposed to a single ANN?

- Such network and comparison has been made but authors decided not to present such single ANN, just mentioned about it in the text. Also presentation of 48 substances ANN would be hard to follow due to large number of data.

How was the decision tree created? I.e. how it was decided which individual classes would be placed into group 1 through 3?

- The process of creation of decision tree was described in the manuscript as follows: "It was achieved after many trials of matching substances, which were not well separated, into new groups and checking if they are good enough on ROC graphs. Consequently, this procedure was also applied to those new groups." - New groups had been tested by creating for them new ANN's and checking by ROC graphs which one separates substances better. Many of them had been trained before the best ones were found. The Final ANN's were learned after dozens of trials.

The authors have indicated on line 203 that the hyper-parameters of the ANNs have been randomly selected until the desired/best result is reached. In terms of reproducibility, it would be helpful to specify the range of parameters which were tested and which of these options produced the best results. Also did each of the 22 networks utilise the same hyper parameters, or was this optimisation conducted for each of the 22 networks?

- It is impossible to reproduce learning process. Even if exactly the same parameters are chosen the learning process will generate each time different result according to randomly chosen initial weights. The range of parameters is typical for backpropagation algorithm and is well documented in the literature. Therefore, authors decided to perform random parameters procedure demonstrated in the paper.

There is no discussion on either data or software availability. The authors need to consider the default Copernicus publishing rules and provide text that would allow others to request access to both the data and software. If this is restricted, it should be stated with the reasons why. https://www.atmospheric-measurement-techniques.net/about/data_policy.html

- The following sentence was added in the manuscript "The experimental aerosol data can be provided upon request. The software for automatic data analysis cannot be commonly provided at this moment since it is a subject of negotiations with a company."

---

## Author Response (AR2)

**Responses to Reviewer's questions**

Our grateful thanks to the Reviewer for valuable comments and thorough analysis of the work, which we believe has really enhanced the manuscript. Our responses are presented below. The deep language correction of the manuscript was also performed by native.

Suggestions for revision or reasons for rejection (will be published if the paper is accepted for final publication)

This manuscript discusses the application of artificial neural networking (ANN) techniques to the previously developed BARDet fluorescence detection system, as well as the potential application to aerosol characterization. In this study, the authors aerosolized 48 different fluorescent aerosols and attempted to model a system of artificial neural networks into a decision tree to appreciably categorize the measured particles. The resulting system was 22 sets of ANNs to totally classify the overall data set over multiple iterations. Real-time and inexpensive bioaerosol classification is an extremely important step to understanding the overall effects bioaerosols have on the environment and climate, and so a paper addressing such could be of great interest AMT community. However, there are some issues here that need to be addressed prior to considering acceptance.

Major

1. Leaving out non-fluorescent particles may be a bigger challenge to successfully implementing the ANNs than the authors suggest, considering that the majority of atmospheric aerosols are largely "non-fluorescent." There's no justification of this further in the text of this manuscript other than the statement in line 195 regarding application of a threshold. Separating a host of fluorescence particle types is one thing, but an atmospheric sample is going to contain an extreme minority of fluorescent particles. A recent study (Savage et al., 2018) utilized HAC techniques to attempt to classify similar types of particles, though the high-thresholding needed to get rid of the majority of the "non-fluorescent" particles ultimately confounded the clustering algorithm due to an appreciable number of "fluorescent" particles being removed as well. If this paper is going to move forward, this needs to be addressed as a limitation of the study. For example, I suggest discussing that nonfluorescent particles being absent is a limitation for usage in ambient studies, though future work could include them in attempts to mimic ambient conditions.

- We agree with Reviewer and we are aware that non-fluorescent particles are dominant in the atmosphere, therefore, their characterization is very important and should not be neglected. However, there is no single technique capable of fully characterizing the entire population of particles in the ambient air. The UV-LIF detectors are intended mainly for detection of biological particles basing on their fluorescence properties. Alternatively size or/and shape of the particles can be measured. There is still no information concerning the bio/chemical composition of measured particles. In our opinion the ANNs alone does not limit "non-fluorescent" particle analysis. They can analyze any type of data and the final result depends on the number, quality and physicochemical parameters which will be used as an input data.

- Regarding the line 195 (currently 122) we justified the types of aerosols tested not the algorithm of data preprocessing. Savage et al. (2018) used intensity thresholds to "extract" stronger signals from those of lower ones. In our experiments all (low and high) fluorescence signals has been considered, but the particles which are non-fluorescent by nature like gypsum, soot, Syloid etc. has not been investigated.

- In the Introduction section in lines 58-60 we added as follows: " Besides advantages such as reagentless and real time particle characterization, the laser based methods do not provide information on the chemical composition of aerosol.. "

- In the last sentence of the Summary we added as follows:  "… which will be most challenging due to the presence of unknown fluorescent and non-fluorescent particles ."

- Of course in real atmospheric studies, that are already planned, the non-fluorescent particles will be considered.

2. The section describing the ANN generation and decision tree processing needs to clarify that this process isn't replicable in terms of the exact factors used for ANN generation, and that the ANN decision-tree generation results may be different with subsequent trials. A response to reviewer 3 from the first submission discusses this in length (ie how the weights/start factors are randomly chosen). I understand this was a real-time attempt as classification, but specific factors being non-replicable as well as having no secondary decision-tree development trials shown, and utilizing a new type of instrument only available to these researchers, can greatly limit possible impacts of this manuscript. This is only compounded by the first point (No non-fluorescent particles probed) in that what was done here may not be applicable to ambient data sets.

- In the section 4.3 in line 371 the following sentence was inserted: " The process of creating them is not replicable in terms of the exact factors used for ANN generation. However, this is not essential, because the decision tree is based on ANN results (classification ability), which should be possibly the highest. Therefore, the final result will be the same."
- We agree with the Reviewer's opinion that instrument and developed software has limited usage by other researchers. On the other hand it is prototype that is still developed, but it potentially may be available commercially in a few years. It is worth to note that for example WIBS family instruments has not been available to other researchers. They are still unavailable for us due to the price. From the practical point of view, it is essentially important to share our experience concerning the new approach to use ANNs to real-time aerosol classification. In the next step we are going to undertake a challenge of real atmospheric data analysis to test the real applicability of the system.

3. The sizes of individual particle types, as well as asymmetry factor, are listed in table 2 for each particle type. It's not clear from the text that these parameters are being used in the ANNs in any way, and in fact the opposite seems to be the case. In terms of the sizes of particles, there are three sub-points:

- The particle sizes and asymmetry factors were proposed by other Reviewer, who suggested the full characterization of the aerosols. Actually those data were not used for aerosol analysis. For ANN analysis only the fluorescence data was used. Numerous trials has shown that best classification was achieved using fluorescence data only.

a. Some of these pollen particles are seen around 85 microns (A. alba), and relatively small particles around 2 microns (Riboflavin) are also being measured. Other commercial UV-LIF instrumentation have issues with detecting simultaneously small and large particles without having limit of detection or saturation issues respectively. The 2016 paper shows some information on size dependence, but only goes up to 8 microns. A statement about the dynamic range of the instrumentation would be helpful here.

- We are aware that commercial devices has been tested in more detail than BARDet which was developed recently and still it is modified and improved. The instrument has been calibrated for particle sizing up to 8 microns because of limited availability of standards in the project. The further research is beyond the scope of the recent grant.
- Regarding the simultaneous detection of low and strong signals we applied entirely new approach. The commercial UV-LIF detectors acquire integrated fluorescence signals in 1-3 fluorescence bands that is likely to produce saturated signals in single window PMT. In the BARDet the fluorescence signal is "distributed" among 18 channels grouped in 7 spectral bands. It assures simultaneously high S/N ratio due to summing low signals as well as prevents the signal saturation by narrow band entering the single PMT channel. It is clearly presented by non-saturated fluorescence characteristics of highly fluorescent riboflavin or FM7 microspheres (Fig. 2).
- In lines 104-106 we added sentence as follows: " Such a solution extends the dynamic range of measured spectra and, assures a high S/N ratio, and also reduces the possibility of signal saturation. ."

b. The average and standard deviation of size is mentioned in this chart, though with no units attached. This needs to be addressed on the table. With the FM7 measurements listed, it appears to be in microns. It is unlikely that the authors would be measuring intact pollen grains with such aggressive sampling methods (intense vortexing/vibration). Aerosolization of pollen has been seen to rupture pollen in previous studies (Hernandez et al., 2016; Savage et al., 2017) as well, let alone aggressive vortexing/vibration. The low uncertainty on the measurements (e.g. 44.8 + 2.01 for S. cereale pollen) also points to intact pollen being measured.

- The missing data concerning particle size units in Table 2 has been completed with "um".

- The data concerning particle sizes in Table 2 are acquired after aerosolization of the samples and collecting them on glass slides. The method we applied was developed especially for non-disruptive aerosol generation. In the experiments gentle vortexing and low air flows has been applied. We agree with Reviewer that pollen fragmentation occurs. However, in the nature pollen grains are resistant to harsh environmental conditions. We checked again the microscopic image and in fact S. cereale grains were intact at least at chosen frame. We do not insist that some of particles were not fragmented. In some cases it was difficult to find representative frame due to low concentration of particles. Moreover, we have noticed that the pollen rupture can occur also inside of the device's nozzle, therefore we think that our aerosol generation method is not the main reason of pollen rupture.

c. Why was only the normalized spectral shape used in the ANN decision making? In a particularly bad example buckwheat flour and cellulose were effectively unable to be classified against one another, though these two particles types showed very different average size and asymmetry factors.

- As it was mentioned earlier there is no single method allowing complete aerosol analysis including size, shape, fluorescence, biological or chemical composition. The advantage of LIF based devices is real-time and no sample preparation analysis. Various naturally occurring biological particles of different origin will fluoresce in similar pattern therefore they will be difficult to distinguish. The numerous approaches has been made concerning input data normalization and selecting which input data should be considered (fluorescence only, fluorescence + scattering). The best results were achieved for fluorescence which is always unchanged in character, while the scattering strongly depends on particle position in interrogation point. The data normalization is essential for proper data calculation and comparison.

4. The paragraph beginning on line 53, describing fluorescent particles and their detection/characterization, seems to be missing several key papers, as well as cites a paper (Hernandez et al., 2016) that is irrelevant to the discussion there. Pan et al., 2007; Crawford et al., 2015; Ruske et al. 2017, 2018; Savage et al., 2017 and 2018 are all examples of recent work that support recent work in the area discussed in the referenced sentence.

- We agree with Reviewer that one of the paper is not a best choice at this point. According to the Rewiever's suggestion the article by Crawford at al. 2015 and Savage et al 2017 has been added.

5. There needs to be mention of the absence of nonfluorescent particles in the abstract, and that further work would need to probe this.

- In the abstract we added last paragraph as follows: " In the future, it is planned that performance of the system may be determined under real environmental conditions, involving characterization of fluorescent and non-fluorescent particles ."

6. Usage of the word "real-time" in the abstract is misleading, because while the instrument does measure in real time the data was collected separately (per aerosol type). The time-component for this study is irrelevant in this case.

- It is true that database collection was separate process. However, as soon as the aerosols library was implemented to the algorithm the next measurements automatically generated immediate aerosol recognition. The term "real-time" relates to measurement of aerosol "fluorescence fingerprints" as well as the data analysis. In this point we refer to supplementary material attached to our article Kaliszewski et al., 2016 which clearly visualizes real-time PCA. The ANN are not so easy to visualize that's why we show final results.

*Minor (or Technical) Points*

1. Raw number of particles per aerosolized particle type is not listed here, and instead a raw number total (114779) for the entire data set is listed, with an average spectra per total listed (~2400). This needs to be addressed, as the statement of 2400 average could be true, though it could also be misleading.

- Actually the average number of spectra can be misleading. In the table statistical data of measurements are presented.

| | |
|---|---|
| all particles for 48 substances | 114751 |
| min | 1548 |
| max | 3609 |
| average | 2391 |
| standard deviation | 437 |

- The sentence in the line 216 has been changed to following: " Finally, a total of 114,779 spectral characteristics of 48 aerosols was gathered, which gives on average 2391 (standard deviation 437) fluorescence characteristics per substance ."

2. Take out the word "impressive" on line 30 (before effectiveness)

- The sentence actually sounds as follows: "As a result, a very high accuracy of aerosol classification in real-time was achieved."

3. Line 119: Leaf scraps should be "leaf litter"

- After reconsideration and literature analysis we assume that "Plant debris" is most suitable term.

4. Naming of things in Table 2 not consistent (e.g. pollen types - some scientific name, some common name), nor are the abbreviations (e.g. Ambio vs FM7 vs PF) which is distracting to the overall data.

- The pollen types in Table 2 has been updated with scientific and common names. The abbreviations were changed as follows: Ambio to AMB, Cin to CIN, Rib to RIB, FM7 to FM. All those abbreviations in the text and in the figures has been updated. The abbreviations Trp and Phe are standard in internationally recognized for aminoacids and remained unchanged.

5. Graph styles are not consistent (Figure 7 and 8 ROC graphs have different tick numbering, as well as background line densities) which is distracting to the overall presentation of the data.
- The tick numbering has been unified.

6. The text size on certain figures (8 and 9) need to be increased, as well as the ROC graphs are low-resolution compared to the confusion matrices listed.

- The text sizes in graphs has been increased from 14 to 18 points. All the figures are in 300 dpi resolution.

7. Line 72: "The simple statistics" isn't the correct syntax. Maybe "simpler statistical analysis"

- The sentence was corrected.

8. Line 119: This line makes no sense currently, and needs reworked.

- The sentence was corrected as follows:  "Due to technical limitations, samples other than pharmaceutical could not be aerosolized in this study".

9. Line 194: "The non-fluorescent particles were not a subject of the research since they can be automatically discarded as non-biological applying given fluorescence threshold." This line needs taken out, because it is fundamentally wrong. As addressed above, the non-fluorescent removal via higher thresholding isn't sufficient reasoning to claim they're going to efficiently be removed because this gets rid of a large overall number of particles (Savage et al., 2017, Savage et al., 2018) and can confound HAC clustering, at the very least. This needs to be mentioned, but as a limitation overall of the scope of this paper.

- In the Major suggestions section p. 1of the response to the Reviewer we answered to this comment. The paper bt Savage was citied.

10. Fluoromax Microspheres are cited as the material used for the FM7, though it doesn't cite the particular fluorescent type (usually listed as a color) used.

- We are agree with the Reviewer that the information about the colour of FM7 microspheres is missing. In Table 2 we added information that green fluorescent microspheres were used.

11. Figure 2's usage of 50 spectra-per-type is more confusing than not to how the input data is used for the ANN training. I assume if the reported 2400 spectra had been visualized it would be much busier, but this gives the impression that the training data only used 50 spectra for each aerosol type, which may or may not be the case.

- Presentation of some spectra representing input data was suggested by other Reviewer. We showed 50 example spectra per one graph since it seems to be optimal for clear presentation. Visualization of 2400 spectra does not make a sense and is practically unachievable with our graphical program module. In the figure description we changed: "Normalized 50 subsequent fluorescence characteristics…" to "Example, normalized 50 subsequent fluorescence characteristics…". In line 200 we inserted the sentence: "From the recorded data 80% was used as a training data set and 20% as test data set." We hope that modified description will be less confusing.

12. Line 16: The term "air contamination" is not usually associated with biological particles, unless there is a specific source of contamination like a waste facility or a mold outbreak.

- The term "air contamination" has been changed to "air pollution".

13. Figure 2 significant figures listed are not uniform for all Size and Asymmetry Factor measurements.

- We are not sure how to answer to the Referee's comment referring to the Size/Assymetry Factor. Figure 2 presents example fluorescence characteristics (not Size/Assymetry Factor) of three different substances.

[revised manuscript text omitted]

---

## Author Response (AR3)

[revised manuscript text omitted]

Sformatowana tabela
Usunięto: 7
Usunięto: 7 um
Usunięto: Rib
Usunięto: pollen
Usunięto: pollen
Usunięto: pollen
Usunięto: pollen
Usunięto: pollen
Sformatowano: Czcionka: Kursywa
Usunięto: pollen
Usunięto: pollen
Usunięto: pollen
Usunięto: pollen
Usunięto: pollen
Usunięto: pollen
Usunięto: pollen
Usunięto: pollen
Usunięto: pollen
Usunięto: pollen
Usunięto: pollen

| 22 | RF | Rice flour | 18.22±6.23 | 0.6±0.07 | MELVIT Poland (*RS) | |
| 23 | TF | Tapioca flour | 12.91±3.41 | 0.7±0.06 | COCK BRAND (*RS) | |
| 24 | WF | Wheat flour | 20.57±4.36 | 0.62±0.07 | MELVIT Poland (*RS) | |
| 25 | Trp | Tryptophan | 15.42±8.96 | 0.81±0.08 | Sigma-Aldrich | amino acids and proteins |
| 26 | Phe | Phenylalanine | 10.41±5.31 | 0.73±0.11 | Sigma-Aldrich | |
| 27 | BSA | Bovine Serum Albumin | 63.8±30.49 | 0.43±0.05 | POCH Poland | |
| 28 | OVA | Ovalbumin | 26.45±5.31 | 0.83±0.07 | POCH Poland | |
| 29 | AMB | *Bif. animalis, S. boulardii, S. thermophilus, L. casei, L. bulgaricus* | 27.97±4.42 | 0.84±0.03 | AMBIO Probiotyk, Lab. Galenowe Poland (*P) | bacteria in medium |
| 30 | LCB | *Lactobacillus bulgaricus* | 51.16±19.33 | 0.68±0.08 | LakciBios, ASA Poland (*P) | |
| 31 | LF | *Bifidobacterium animalis, L. acidophilus* | 32.62±8.45 | 0.82±0.07 | Linex forte, LEK Pharmaceuticals d.d. Slovenia (*P) | |
| 32 | BA | Bacteriological Agar | 49.47±10.03 | 0.74±0.07 | Sigma-Aldrich | medium |
| 33 | BAB | Blood Agar Base | 18.78±2.11 | 0.71±0.12 | Sigma-Aldrich | |
| 34 | LB | Luria broth | 15.11±6 | 0.67±0.07 | Sigma-Aldrich | |
| 35 | NB | Nutrient broth | 42.67±9.21 | 0.69±0.03 | Sigma-Aldrich | |
| 36 | BTSTG | *Bacillus thuringiensis* spores technical grade | 7.13±5.95 | 0.72±0.12 | Agricultural | Bacterial spore with admixtures |
| 37 | SB | *Saccharomyces boulardii* | 57.82±7.56 | 0.69±0.05 | Enterol, Biocodex France (*P) | fungi with admixtures |
| 38 | SC | *Saccharomyces cerevisiae* | 21.33±5.55 | 0.76±0.07 | Dr. Oetker Germany (*RS) | |
| 39 | LS | *Lycoperdon* spores | 14.52±0.62 | 0.92±0.02 | (*OC) | fungal spores |
| 40 | JGSS | Johnsons grass smut spores | 6.91±0.34 | 0.98±0.02 | Duke Sci. Corp. | smut spore (fungal spore) |
| 41 | BGSS | Bermuda grass smut spores | 6.47±0.27 | 0.97±0.02 | Duke Sci. Corp. | |
| 42 | ACFTD | AC Fine Test Dust | 3.47±2.34 | 0.87±0.09 | Duke Sci. Corp. | other |
| 43 | NT | Nivea talc | 14.33±4.71 | 0.77±0.09 | Nivea Baby (*RS) | |
| 44 | PPD | Printer paper dust | 76.37±18.89 | 0.43±0.11 | XEROX Laserprint collected from paper shredder (*RS) | |
| 45 | PTD | Paper towel dust | 73.45±25.65 | 0.56±0.15 | Merida Poland collected from crushed towel (*RS) | |
| 46 | CIN | Cinnamon | 23.97±4.39 | 0.78±0.05 | Kamis Poland (*RS) | |
| 47 | CEL | Celulose | 82.86±14.28 | 0.25±0.04 | Sigma-Aldrich | |

Usunięto: Ambio

Sformatowana tabela

Sformatowana tabela

Usunięto: Cin

Usunięto: C

Usunięto: el

[revised manuscript text omitted]

Sformatowano: Odstęp Przed: 0 pkt